# Pressure-Induced Tunable Charge Carrier Dynamics in Mn-Doped CsPbBr_3_ Perovskite

**DOI:** 10.3390/ma15196984

**Published:** 2022-10-08

**Authors:** Luchao Du, Xiaoping Shi, Menghan Duan, Ying Shi

**Affiliations:** Institute of Atomic and Molecular Physics, Jilin University, Changchun 130012, China

**Keywords:** Mn-doped inorganic perovskite, pressure, crystal structure, carrier dynamics

## Abstract

All-inorganic perovskite materials (CsPbX_3_) have attracted increasing attention due to their excellent photoelectric properties and stable physical and chemical properties. The dynamics of charge carriers affect the photoelectric conversion efficiencies of perovskite materials. Regulating carrier dynamics by changing pressure is interesting with respect to revealing the key microphysical processes involved. Here, ultrafast spectroscopy combined with high-pressure diamond anvil cell technology was used to study the generation and transfer of photoinduced carriers of a Mn-doped inorganic perovskite CsPbBr_3_ material under pressure. Three components were obtained and assigned to thermal carrier relaxation, optical phonon–acoustic phonon scattering and Auger recombination. The time constants of the three components changed under the applied pressures. Our experimental results show that pressure can affect the crystal structure of Mn-doped CsPbBr_3_ to regulate carrier dynamics. The use of metal doping not only reduces the content of toxic substances but also improves the photoelectric properties of perovskite materials. We hope that our study can provide dynamic experimental support for the exploration of new photoelectric materials.

## 1. Introduction

Perovskite materials with ABX_3_ structures have excellent properties, such as tunable bandwidth, ultrafast charge-generation speed, high electron–hole mobility and long carrier lifetimes [1,2,3]. In recent years, the photoelectric conversion efficiency of perovskite has been increased from 3.8% to more than 25% [4,5,6,7], the organic–inorganic hybrid lead halide perovskite (CH_3_NH_3_PbI_3_ (MAPbI_3_)) being the first to attract wide attention, mainly due to its strong photoelectric performance [8], Although CH_3_NH_3_PbI_3_ (MAPbI_3_) perovskite material has superior photophysical properties and a very high conversion efficiency, its stability is the key factor restricting its practical application. CH_3_NH_3_PbX_3_ perovskite material easily decomposes in high-temperature and high-humidity environments [9,10], mainly due to the inherent hydrophilicity and distorted tetragonal crystal structure of the material [10], which is the main bottleneck restricting its practical application at present. The goal of many researchers is to try to improve both the stability conditions of perovskite and its photoelectric performance. Scientists have confirmed that replacing the MA^+^ cations with Cs^+^ ions in inorganic perovskite not only improves the stability but also improves the photoelectric performance. At the same time, this experimental operation makes the preparation of perovskite easier [11,12,13,14,15].

All-inorganic perovskite lead halide peroxide nanocrystals (NCs) are used in many scientific fields due to their advantages of a tunable band gap and better stability. Lead halide peroxide nanocrystals (NCs) have bright luminescence and broad spectral tunability and thus show great potential for use in the development of intelligent probes for biological imaging [16]. In recent years, an increasing number of researchers have also focused on the application of all-inorganic perovskite in the photovoltaic field, and metal halide peroxides have been successfully applied in OLED and some photocatalytic reactions due to their wide optical absorption ranges and easily adjustable band gaps [17]. Lead-free MA_3_Bi_2_Br_9_-COFs nanocomposites can even more effectively photocatalyze the polymerization of various functional monomers, which is an important step in exploring the function of oxide photocatalysis, achieving a conversion rate of 97.5% [18]. Although a lot of research has been carried out regarding the development of all-inorganic perovskite materials in various fields, the microscopic physical mechanisms are still unclear, such that it is very important and significant to further understand the relationships between the structure and the physical and chemical properties of perovskite.

The crystal structure plays an important role in determining the fundamental photophysical processes of perovskite materials. In order to further improve the performance of photoelectric devices based on all-inorganic perovskite materials, scientists use various chemical methods to regulate the structure of perovskite materials, such as surface ligand treatment [19,20], changing the synthesis method and the ratios of lead and halogen ions [21], and doping perovskite nanocrystals with metal ions [22,23,24,25,26,27,28,29,30,31]. The properties of matter under high pressure have been widely explored. Pressure can act on the crystal structures of materials and change the correlations of original electrons and the interactions between electrons and lattices [32,33,34,35]. Pressure can also effectively change the electronic arrangement of atoms in matter. For perovskite materials, the separation and transport of carriers are closely related to the crystal structure and other microstructural features. The addition of pressure will change the electronic band structure due to change in the lattice structure, which will lead to changes in carrier excitation states and carrier transmission rates.

The effects of pressure on the structures and properties of perovskite materials have been studied by scientists. The orbitals formed by CsPbBr_3_ perovskite are Pb 6s and Br 4p orbitals and Pb 6p and Br 4p orbitals, respectively [36], and the band gap of CsPbBr_3_ perovskite is 2.32 eV [37]. Karunadasa’s group studied the conductivity of perovskite by changing the pressure of Cu-Cl mixed perovskite. It was found that the structure of perovskite changed with an increase in pressure and that the conductivity of perovskite increased with the increase in pressure [38]. In 2016, Zou’s group [39] studied the optical stability and structural stability of organometallic perovskite crystal FAPbBr_3_ under pressure. With the increase in pressure, the fluorescence color of FAPbBr_3_ crystal gradually changed from translucent orange to red, then to yellow and finally became colorless, and the photoluminescence of perovskite gradually changed from green to yellow and then disappeared. Data obtained with high-pressure synchrotron radiation confirmed that there are two phase transitions (*Pm3m*→*Im3*→*Pnma*) in perovskite, with an initial red shift and subsequent blue shift of spectra under pressure, and it was observed that amorphization occurred at about 4.0 GPa. Four years later, Zou’s group studied the changes in the luminescence properties of Mn-doped perovskite due to pressure. They found that pressure induced the phase transition of the CsPb_x_Mn_1__−__x_Cl_3_ crystal structure, and the maximum photoluminescence intensity under high pressure was significantly enhanced compared to the initial intensity. Energy transfer became more effective with the increase in pressure due to the energy release of Mn increasing from the 4T_1_ state to the 6A_1_ state. The luminescence intensities of CsPb_x_Mn_1−x_Cl_3_ and CsPbCl_3_ perovskite materials are different under pressure. It was further proved that adding Mn elements can obtain better luminescence performance [40]. In 2017, Wang’s group studied the structural changes in perovskite under pressure, and it was found that the first structural phase transition of all-inorganic perovskite occurred with a pressure near 1.0 GPa and that the optical properties of perovskite changed with the increase in pressure. Their study provides an experimental basis for high pressure to modulate the structure and properties of the fully inorganic halide perovskite [37].

Time-resolved spectroscopy is an important scientific and technological means of studying the carrier dynamics of perovskite materials. At present, there is little research on the ultrafast dynamics of metal-ion-doped perovskite under pressure. Studying the ultrafast process of photocarriers of metal-ion-doped all-inorganic perovskite under pressure is helpful for exploring conditions to improve the photoelectric properties of all-inorganic perovskite. In this paper, we will discuss the ultrafast dynamics of Mn-doped inorganic perovskite CsPbBr_3_ under different pressure conditions. The ultrafast carrier dynamics of Mn-doped CsPbBr_3_ under pressure were experimentally investigated by combining transient absorption spectroscopy technology with diamond anvil cell technology [41]. The effects of pressure-induced crystal-structure changes on photogenerated carrier dynamics will be discussed in detail.

## 2. Materials and Methods

### 2.1. Synthesis of Mn-Doped CsPbBr_3_ Nanocrystals

CsPbBr_3_ perovskite nanocrystals were synthesized by a high-temperature thermal injection method [42], the detailed steps of which were as follows: (1) Synthesis of the Cs-oleate precursor solution: Under the condition of inert N_2_ gas, cesium carbonate (0.35 g, Cs_2_CO_3_, Sigma-Aldrich, St. Louis, MO, USA, 99.9%) and octadecene (20 mL, ODE, Sigma-Aldrich, 90%), together with oleic acid (1.25 mL, OA, Sigma-Aldrich, 90%), were put in a 50 mL three-neck round-bottom flask and continuously stirred. At this time, the temperature of the reaction solution was maintained at 150 °C until all the Cs_2_CO_3_ and OA reacted completely, and the resulting transparent solution was stored for standby. (2) The preparation of CsPbBr_3_ nanocrystals: Octadecene (12 mL, ODE, Sigma-Aldrich, 90%) was put in a 50 mL three-neck round-bottom flask, and the solution temperature was controlled at 100 °C. Then, PbBr_2_ (0.1725 g, Sigma-Aldrich, 98%) was added, and the reactant was stirred continuously for 1 h under N_2_ conditions. The reaction temperature was then set to 140 °C, and 1.5 mL of OA and 1.5 mL of oleylamine (OLA, Sigma-Aldrich, 70%) were injected. When the solution became clear, at 140 °C, 1 mL of cesium oleate precursor solution was quickly injected into the reaction mixture. This had to be preheated to 140 °C before injection, because cesium oleate will precipitate from octadecene at room temperature and will appear light green within 5 s. Then, the three-neck flask was immersed in an ice-water mixture for immediate quenching, and a crude solution of CsPbBr_3_ nanocrystals was obtained. Next, the CsPbBr_3_ nanocrystalline crude solution was washed twice. First, methyl acetate (MeOAc) was added to the crude solution at a volume ratio of 1:2, then centrifuged at 4000 rpm for 15 min, and the precipitate was dispersed in 3 mL of n-hexane solvent. MeOAc was added to the obtained solution again at a volume ratio of 1:2, centrifuged at 4000 rpm for 15 min, and the precipitate was dispersed in 5 mL of n-hexane to obtain the purified CsPbBr_3_ nanocrystalline solution. (3) Preparation of Mn-doped CsPbBr_3_ [31]: Under ultrasonic conditions, 0.02 g of MnBr_2_ powder was dissolved in 2 mL of MeOAc to obtain a MnBr_2_ precursor. A quantity of 20 μL of MnBr_2_ precursor was added to 1 mL of CsPbBr_3_ nanocrystals and stirred continuously for 1 min to obtain Mn-doped CsPbBr_3_ nanocrystals. The above solution was then centrifuged at 4000 rpm for 15 min, the precipitate was removed, and 2 mL of MeOAc was added to the supernatant for washing, followed by further centrifugation at 4000 rpm for 15 min and dispersal of the precipitate in 2 mL of hexane to obtain a solution of Mn-doped CsPbBr_3_. The above perovskite samples were naturally dried when the following transient absorption spectrum experiments under pressure were carried out. XRD characterization of manganese-free perovskite was performed in our previous work, by means of which the structure of the crystal was determined to be cubic [43]. The sample preparation was based on the reference experimental preparation method [41,42], and the XRD characterizations of manganese-doped and undoped perovskite showed that the lattice compression of perovskite NCs after Mn doping enhanced the replaceable ability of Pb^2+^ with Mn^2+^ and the structure was unchanged.

### 2.2. Femtosecond Time-Resolved Spectroscopy under Pressure

The pump-probe femtosecond transient absorption spectroscopy used in this work has been described in our previous study [43]. The spectrometer works with an ultrafast (50 fs) amplified titanium–sapphire laser system (Libra-USP-HE, Coherent Inc., Santa Clara, CA, USA) with a center wavelength of 800 nm and a repetition rate of 1 kHz. In the in situ high-pressure transient absorption spectroscopy experiments, the equipment used for pressure generation was a diamond anvil cell (DAC) with an anvil culet 800 μm in diameter [44]. We used a laser drilling machine to pre-indent the T301 stainless steel into a thickness of 100 μm, and then drilled a cavity (about 500 μm in diameter) at the center of the indentation. A ruby with a diameter of about 5 μm was placed in the sample cavity as the standard pressure material. When the experimental samples were loaded into the press, the CsPbBr_3_ NCs and ruby were put into the sample chamber together, and silicone oil was used as the pressure transfer medium for the sample, because silicone oil only acts as a suspension; it does not affect the nature of the sample itself [45]. It needs to be noted that only one ruby needs to be placed in the sample cavity to avoid the inaccurate absorption peak of ruby displayed when the press is pressurized. The pressure is increased by twisting the rotating screw on the diamond on top of the press. The actual pressure can be determined by standard ruby fluorescence techniques. In the process of pressure calibration, the R1 fluorescence peak of a ruby ball was chosen as a reference. A hydrostatic pressure condition was obtained by using silicon oil as the pressure-transmitting medium, which avoided the introduction of impurity additional spectra to those of the sample. All operations of pressurization were performed under the microscope. Figure 1 shows an optical path diagram of femtosecond transient absorption spectroscopy under pressure. During the experiment, our laser intensity was controlled below 1 mw to avoid damaging the ruby. For the accuracy of the experiment, we basically carried out the experiment at a constant temperature and humidity.

## 3. Results and Discussion

### 3.1. Optical Properties of CsPbBr_3_

Before studying the pressure effect, a basic understanding of the optical properties of CsPbBr_3_ in ambient conditions is necessary. The steady-state absorption spectra of CsPbBr_3_ and Mn-doped CsPbBr_3_ particles dispersed in silicon oil were measured in ambient conditions. Figure 2 shows the steady-state absorption spectra normalized at the peak values for CsPbBr_3_ and Mn-doped CsPbBr_3_ collected by a V-670 (JASCO, Tokyo, Japan) spectrometer under normal pressure. The proportion or type of halogen anions in all-inorganic perovskite will affect the peak position of its absorption peak. For all-inorganic perovskite without Mn-ion doping, its absorption peak is at 500 nm [46]. In our experiment, the peak position of the absorption spectrum of Mn-doped CsPbBr_3_ perovskite at normal pressure was at 512 nm, showing a red shift. After Mn doping, the internal structure of perovskite changed, indicating that the red shift of the absorption peak of Mn-doped CsPbBr_3_ can be adjusted by changing the composition of perovskite. The doping with Mn metal ions led to the red shift of the absorption peak shown in Figure 2, mainly because doping with metal ions leads to a change in the band gap of perovskite [47]. In addition, it was confirmed that Mn ions were doped into the CsPbBr_3_ sucessfully.

### 3.2. Transient Absorption Spectra of Mn-Doped CsPbBr_3_ at Atmospheric Pressure

We measured the transient absorption spectra of Mn-doped CsPbBr_3_ at different decay times to characterize their photoexcited-state properties under atmospheric pressure, as shown in Figure 3. Femtosecond transient absorption experiments were carried out under the excitation of a 400 nm laser wavelength. ΔOD represents the change in optical density. The obtained spectra shown in Figure 3 cover the wavelength range from 450 nm to 630 nm. There are two characteristic peaks in Figure 3, namely, PA1 and PB1. The positive transient absorption band PA1 was ascribed to the signal of excited-state carriers [48,49,50]. The signal of ground-state bleaching PB2 was centered at around 512 nm, which coincided with the bandgap obtained in our steady-state spectra; it takes nearly 2 ps for the bleaching peaks to reach its maximum intensity. We analyzed the transient absorption spectrum of Mn-doped CsPbBr_3_ using global fitting (Figure 4 shows the decay correlation spectrum), and three time constants, 565.0 ± 19 fs, 28.15 ± 1.0 ps and 246.6 ± 15 ps, were obtained. The time scale obtained in our study was very similar to that for the dynamics of carriers in MAPbBr_3_ film measured by Deng et al. [51]. In 2017, Haiming Zhu and others explored the ultrafast behaviors of carriers in three perovskite crystals with different cations using time-resolved fluorescence spectroscopy and found that the change of cation type in perovskite does not affect the ultrafast dynamics of carriers [52]. For the origin of the transient components in Figure 4, we attribute 565.0 ± 19 fs to the relaxation process of hot carriers [53,54] and 28.15 ± 1.0 ps is the scattering process of optical phonons and acoustic phonons [51]; the process with a time scale of 246.6 ± 15 ps is Auger recombination [55]. Auger recombination is an inverse process of collision ionization, and it is also a non-radioactive electron–hole recombination process, that is, electrons at high energy levels are directly combined with holes to transfer energy to another carrier. This recombination method does not emit light but releases energy in the form of phonons.

### 3.3. Transient Absorption Spectroscopic Analysis of Mn-Doped All-Inorganic Perovskite CsPbBr_3_ under Different Pressures

In order to explore the ultrafast process of Mn-doped CsPbBr_3_ under the effect of pressure, femtosecond pump-probe technology was used to measure the transient absorption spectra of Mn-doped CsPbBr_3_ under different pressure conditions. Pressure was applied using diamond anvil cell technology. Figure 5a,b shows the normalized transient absorption spectra of Mn-doped and undoped perovskite CsPbBr_3_ under different pressures (from atmospheric pressure to 1.4 GPa) with the same delay time scale (10 ps). The maximum applied pressure was 1.4 Gpa, because the signal of the sample was very weak and the signal-to-noise ratio was very poor under higher pressures, so it was not easy to obtain high-quality experimental data. From Figure 5a, it can be seen that the bleach signal of the transient absorption spectra of Mn-doped CsPbBr_3_ was centered at around 515 nm when the pressure was 0 Gpa and that the bleaching peak of the transient absorption spectra shifted blue to 496 nm when the pressure increased to 0.2 Gpa; as the pressure continued to increase, the bleaching peak started red-shifting compared to that at 0.2 Gpa as the pressure continued to increase to 1.2 Gpa, and the bleaching peak was around 525 nm. When the pressure increased to 1.4 Gpa, the blue shift of the bleaching peak began to occur again. Similarly, we also found in Figure 5b that in the CsPbBr_3_ system without Mn ions, with the same applied pressure, the bleaching peak position also appeared blue-shifted first, then red-shifted and then blue-shifted. However, the applied pressure corresponding to the shift in peak position was different, mainly due to the doping with manganese ions. Zou’s group [40,45] found that all-inorganic perovskite materials with and without Mn doping were excellent metal halide perovskites with high pressure resistance, such that metal halide perovskites are suitable materials for use in high-performance perovskite solar cells and optical pressure sensors. They found that the photoluminescence of CsPb_x_Mn_1__−x_Cl_3_ NCs can persist even at 20 Gpa. In our experiment, changing the pressure up to 1.4 Gpa only caused the phase transition of the crystal structure without degrading the perovskite crystal. We suggest that pressure has a bidirectional adjustment effect on the band gap of perovskite CsPbBr_3_ doped with Mn ions. In order to better understand the relationship between the shift in bleaching peak position and pressure, the evolution of the transient absorption peak energy-level change in Mn-doped and undoped perovskite as a function of pressure is shown in Figure 6 at a 10 ps time delay. As the pressure increased from 0 Gpa to 0.2 Gpa, the band gap, expressed in electron volts (ev), changed from 2.41 ev to 2.49 ev; when the applied pressure continued to increase from 0.2 Gpa to 1.2 Gpa, the narrowing of the band gap was from 2.49 ev to 2.36 ev. When we continued to increase the pressure from 1.2 Gpa to 1.4 Gpa, the band gap moved from 2.36 ev to 2.38 ev.

Figure 1 shows a schematic diagram of the orbital hybridization of the conduction band minimum (CBM) and valance band maximum (VBM) of Mn-doped CsPbBr_3_. In Figure 1, it is suggested that the change in the structure of [PbBr_6_]^4−^ has a great impact on the band gap of Mn-doped CsPbBr_3_. We think the bi-directional adjustment of the band gap of Mn-doped CsPbBr_3_ should be due to the change in the octahedral configuration of [PbBr_6_]^4−^ under the application of pressure [56,57]. In detail, the CBM is relatively insensitive to bond length or pressure [56] because it is mostly a nonbonding localized state of Pb p orbitals. With the increase in applied pressure, the change in bond length and bond angle of perovskite has no obvious effect on CBM. On the contrary, the change in bond length and bond angle of perovskite has a great impact on the VBM. Under the condition of adding relatively mild pressure to a sample at the beginning, the Pb-Br-Pb bond angle remains unchanged, the atomic spacing of perovskite decreases due to the shortening of bonding length (Pb-Br bond), the interaction between Pb 6s and Br 4p orbitals is enhanced and the rise in VBM leads to the narrowing of the band gap. However, with increasing pressure on perovskite, the structure of [PbBr_6_]^4−^ octahedra tilt gradually, the crystal structure changes, appearing as the bond angle of Pb-Br-Pb deviates from a straight angle, and the band gap will increase due to weakened orbital coupling.

In our experiment, the band gap of Mn-doped CsPbBr_3_ decreased when the applied pressure changed from 0.2 GPa to 1.2 GPa (Figure 6). When the pressure was greater than 1.2 GPa, the reduction in the bond angle of perovskite led to a decrease in VBM and a reverse increase in band gap. Therefore, the absorption peak in our measured transient absorption spectrum showed a trend of blue shift first and then red shift with the increase in pressure [56,57]. In order to determine clearly the effect of pressure on the dynamics of photogenerated carriers in Mn-doped CsPbBr_3_ perovskite, global fitting was performed on the transient absorption spectra of Mn-doped CsPbBr_3_ under different pressures. Three time constants were obtained. In order to better understand the influence of pressure on carrier dynamics, the changes in the time constants of the three components as a function of pressure are shown in Figure 7. Figure 8 shows the dynamic traces extracted from the transient absorption spectra of Mn-doped CsPbBr_3_ under 0.2 GPa, 0.58 GPa and 1.2 GPa by global fitting. As shown in Figure 7a–c, the time constants of the three components are marked as τ_1_, τ_2_ and τ_3_. We compared the carrier dynamics of perovskite with and without Mn doping under different pressures. From Figure 7a, the τ_1_ of perovskite doped with Mn ions under different pressure was smaller than that of perovskite without Mn ions on the whole, because the radii of Mn ions are relatively short. With other conditions unchanged, doping with Mn ions leads to a reduction in the crystal spacing of perovskite and the enhancement of lattice interactions. τ_1_ was assigned to the relaxation process of hot carriers; the rate of the relaxation process of hot carriers is faster for Mn-doped perovskite. For perovskite materials doped with metal ions, the Pb-Br bond length is shortened with an increase in pressure, which has little effect on the CBM, while the increase in its VBM is attributed to the compression of Pb-Br bond length, which reduces the band gap of perovskite doped with metal ions. In the process of increasing pressure, under the action of a certain degree of stress, the crystal structures of perovskite materials will undergo different degrees of deformation. When the atomic spacing is reduced, the ability to resist deformation is stronger. When the pressure increased, the time constant of hot carrier relaxation, τ_1_, decreased due to the enhanced interaction between carriers and phonons.

τ_2_ belongs to the scattering process of optical phonons and acoustic phonons [51]. From Figure 7b, we found that, under the same pressure, the τ_2_ of CsPbBr_3_ perovskite doped with Mn ions was larger than that of CsPbBr_3_ undoped with Mn ions on the whole. The phase-transition points of the two systems are different. Compared with the applied external force, the stress caused by the doping with Mn ions is very small, resulting in the phase-transition points of Mn-ion-doped and undoped perovskite being different under similar pressures. In Figure 7b, it can be seen that the phase transition of CsPbBr_3_ occurred from 0.2 GPa to 0.81 GPa, while the phase transition of Mn-doped CsPbBr_3_ perovskite occurred from 0.2 GPa to 1.2 GPa. The experimental data show that the phase-transition point of CsPbBr_3_ doped with Mn ions was higher than that of CsPbBr_3_ without Mn-ion doping. Under the same pressure, the time constant τ_2_ of optical phonon and acoustic phonon scattering of CsPbBr_3_ doped with Mn ions was longer than that of CsPbBr_3_. We think that after doping with Mn ions, the atomic radii of Mn atoms are shorter than those of Pb atoms and that the band gap of perovskite will be further reduced, hence, the coupling between optical phonons and acoustic phonons will be enhanced. The lattice compression of perovskite doped with Mn ions is greater than that of perovskite without Mn ions under pressure, so CsPbBr_3_ doped with Mn ions is more stable and will produce better photoelectric properties under pressure [40]. The proposed mechanism is as follows: with a gradual increase in pressure, the energy is transferred from the excited state of perovskite to the 4T_1_ state of Mn ions, which reduces the energy of optical phonon–acoustic phonon scattering, slows down the thermal equilibrium and then increases the coupling time. When the pressure continues to increase to 1.2 GPa, structural phase transition occurs. The radiation energy of CsPbBr_3_ doped with Mn ions from the excited state to the ground state is suppressed. Therefore, τ_2_ became larger with the increase in pressure before its transformation. We found that when the pressure increased to 1.2 GPa, the time constant of τ_2_ decreased significantly and was much smaller than that under atmospheric pressure. We speculate that this is due to the severe distortion of octahedra, which leads to disruption of the lattice order.

The third process belongs to Auger recombination. From Figure 7c, it can be seen that τ_3_ underwent an obvious change at the pressures of 0.2 GPa and 1.2 GPa. Liu et al. found that the phase transition of MAPbBr_3_ perovskite occurred at a pressure of 0.4 GPa and 1.2 GPa with the increase in pressure. The critical pressure was related to the change in microcarrier dynamics and macroelectrical properties [58]. From the three stages of change in τ_3_, the three different phases of Mn-doped CsPbBr_3_ perovskite should be cubic, tetragonal and orthogonal phases [59,60]. Among the three phases, the tetragonal phase has the best structural symmetry, so the rate of Auger recombination is accelerated. The trap state of perovskite changes with the increase in pressure and the energy is also transferred from the excited state of perovskite to the 4T_1_ state of Mn ions, thus, it promotes the transfer of carriers to the trapped state, and then the Auger recombination process occurs after the electrons are captured in the trapped state. With the increase in pressure, the energy transfer accelerates, which promotes the possibility of being trapped, thus accelerating the Auger recombination process. It can be seen from Figure 7c that the τ_3_ of perovskite doped with Mn ions was slower than that of perovskite without Mn-ion doping. We believe that doping with Mn ions leads to an increase in the defect states of perovskite, that is, it inhibits the charge recombination process, and the lifetime of the charge recombination process of perovskite doped with Mn ions became longer. After the introduction of pressure-induced phase transition in CsPbBr_3_ perovskite, the structural phase transition will enhance the dispersion of various energy states in the energy band, which is also conducive to meeting the needs of particle-energy and momentum conservation in the Auger recombination process, accelerating the reaction speed of the process, the time constant decreasing with the increase in pressure [61,62]. In our study, the ratio of manganese to lead was 4.5 to 100. We think that the Mn ratio will have had an influence on the results. We chose one manganese doping ratio because there would be no way of explaining the mechanism if many factors were changed at the same time. The influence of pressures on the carrier dynamics of perovskite doped with different ratios of manganese is also very interesting. We look forward to further explorations in future research.

## 4. Conclusions

Transient absorption spectra of Mn-ion-doped and undoped all-inorganic perovskite CsPbBr_3_ at different pressure ranges were obtained. Three pressure-dependent time constants, τ_1_, τ_2_ and τ_3_, were obtained, which can be classified as: hot carrier relaxation, optical phonon–acoustic phonon scattering and Auger recombination. We elucidated the physical mechanism governing how the ultrafast kinetic process of photocarriers of Mn-ion-doped perovskite is regulated by pressure. The experimental results showed that increasing pressure has a bidirectional effect on the band gaps of the above perovskites, indicating that pressure changes the crystal structure of perovskite, causing it to undergo phase transformation. The phase transition of the CsPbBr_3_ occurred at the pressure of 0.81 GPa, while the phase transition point of Mn-doped CsPbBr_3_ occurred at a pressure of 1.2 GPa. It can be seen that the external pressure required for the phase transition of Mn-ion-doped perovskite was higher. The lattice compression energy of the perovskite CsPbBr_3_ doped with Mn ions was higher than that of CsPbBr_3_. Doping with Mn ions makes perovskite more stable under pressure. Our work is expected to provide experimental support for the discovery of more stable perovskite materials.

## Data Availability

Data are contained within the article.

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
