# Peer review of "Pressure-Induced Tunable Charge Carrier Dynamics in Mn-Doped CsPbBr3 Perovskite"

_materials, 2022, doi:10.3390/ma15196984_

Round 1

Reviewer 1 Report

The authors studied the ultrafast dynamics of Mn-doped inorganic perovskite CsPbBr3 under different pressure conditions. The work is well performed, and paper is well written. The novelty is high as well. However, there are still some problems in the experiments, data presented, references, introduction, and figure quality. Therefore, the following questions must be addressed.

1.  Some figures have low quality. For example, Figure 1, the y axis is better started with 0 instead of -0.1.  Figure 7 is too small to look at.  Figure 4 a and b are not aligned well. Figure 6, the a, b, c should be outside of the figure.

2. What is “OD” in the figure axis should be noted.

3. Figure 1, please normalize the absorption spectrum. Besides, use “intensity” might be better than “OD”.

4. The basic characterization of perovskites is lacking. I understand that the aim of this paper is more about time resolved spectroscopy, but at least the XRD should be provided to prove the formation of right perovskite structure.

5. The introduction should be expanded and more discussion about the advantage/ applications of perovskites should be added. In this way, the significance of why study this material and the paper could be improved. For example, halide perovskites have been widely used in bioimaging (https://doi.org/10.1002/adfm.201604382), OLED and photocatalysis (https://doi.org/10.1021/acsmacrolett.0c00232; https://doi.org/10.1021/acsmaterialslett.1c00785), and the above related paper could be cited.

6. Does the authors know the Mn ratio? Would the Mn ratio have influence on the results? The authors can at least add some discussion/outlook.

7.  How did authors add pressure to pump-probe system is not clear to me. The perovskites are in solution system, as the authors described. With the pressure change, will it also influence the solvent and not only the perovskites?  Besides, adding a scheme about the set-up could be better.

8. One major problem the authors did not explore is whether the perovskites would degrade under high pressure. If so, some of the authors’ conclusion would be challenged.

9.  Figure 4, the data points measured at 1.4 GPa is clearly at somewhere over 1.4 GPa.

Reviewer 2 Report

The manuscript "Pressure-Induced Tunable Electron Transfer Rates in Mn3 doped CsPbBr3 Perovskite" presented by Luchao Du et al. investigates how transient absorption (TA) spectra and kinetics of the perovskite depend on external pressure. The authors determined the decay-correlated TA spectra for undoped material and assigned them to different stages of energy dissipation in excited perovskite. After this, similar analysis was performed for Mn-doped material. Quite expectedly is that the decay characteristics of the Mn-doped perovskite and their dependence on pressure are different from those in undoped material. This part of the work looks rather well. However, the part concerning with electron transfer with participation of Mn ion is rather speculative and is not supported by any experimental data. Because of this, I do not agree that the term "electron transfer rate" is included in the title of the manuscript: the authors simply observe and describe TA parameters, to use the term in title, special investigations are required confirming that this is really electron transfer. In general, what they observe is the excitation energy relaxation process, the term "electron transfer" is different, it means that the process is irreversible.
And of course, the level of English should be improved essentially, at the moment it is inacceptable for publication.

Round 2

Reviewer 1 Report

The authors did a wonderful job in revision 

The current version should be accepted